LETTER TO THE EDITOR

# Performance of the GeneSoC Rapid PCR System in Detection of SARS-CoV-2 from Saliva Specimens

Kenji Ota,[a] Ryo Kurahara,[a] Chie Tsukamoto,[a] Yasuhide Kawamoto,[a] Norihiko Akamatsu,[a] Daisuke Sasaki,[a] Fujiko Mitsumoto-Kaseida,[a] Kei Sakamoto,[a] Kosuke Kosai,[a] Hiroo Hasegawa,[a] Kazuko Yamamoto,[b*] Koichi Izumikawa,[c] Hiroshi Mukae,[b] Katsunori Yanagihara[a]

[a]Department of Laboratory Medicine, Nagasaki University Hospital, Nagasaki, Japan
[b]Department of Respiratory Medicine, Nagasaki University Hospital, Nagasaki, Japan
[c]Infection Control and Education Center, Nagasaki University Hospital, Nagasaki, Japan

**KEYWORDS** SARS-CoV-2, COVID-19, saliva, GeneSoC

Since the beginning of the coronavirus disease 2019 (COVID-19) outbreak, the detection of SARS-CoV-2 by nucleic acid amplification tests, including reverse transcription (RT)-PCR, has been the gold standard for diagnosis (1, 2). Rapid and accurate detection of SARS-CoV-2 is of great importance, nevertheless, RT-PCR requires several hours of turnaround time (TAT), attributable to the nucleic acid extraction (sample preparation) and amplification processes. GeneSoC (Kyorin Pharmaceutical Co., Ltd., Tokyo, Japan) is a rapid PCR system that detects SARS-CoV-2 RNA (N2 region) sequences in less than 15 min using microfluidic thermal cycling technology requiring simple procedures (3). The performance of SARS-CoV-2 RNA detection has been reported for nasopharyngeal swab specimens (4, 5). In the COVID-19 era, saliva is another commonly used specimen for diagnosing COVID-19 with comparable sensitivity and specificity to nasopharyngeal swabs (6–8). However, collection, storage, and processing methods for saliva specimens are not standardized sufficiently (9), so the study comparing saliva and other types of specimens needs careful interpretation before clinical application. In addition, sample preparation, including nucleic acid extraction and purification, is another time-consuming process for using saliva as test specimen. Therefore, individual evaluation for SARS-CoV-2 detection method in saliva is necessary. GeneSoC Lysis Buffer (GLB) is a saliva sample preparation kit that enables saliva sample preparation (nucleic acid extraction) in several minutes. The objective of this study is clarifying the performance of GLB and the GeneSoC N2 detection kit (N2 kit) using clinical saliva samples (Institutional Review Board approval number, 21051706-2).

Clinically obtained and stored (−80°C) saliva samples were used in this study. These samples were obtained from February 2022 to March 2022, when omicron variants were dominant. Saliva specimens were self-collected by spitting directly into a sterile tube, with fasting prior 30 min. The samples had been stored at 4°C before transported to our laboratory. To evaluate the performance of the N2 kit, nucleic acid was extracted and purified from saliva using the MagMAX Viral/Pathogen Nucleic Acid isolation kit by Thermo Fisher Scientific (MVP) following the manufacturer's protocol. Briefly, nucleic acid in saliva specimens (400 $\mu$L) were purified and condensed in 100 $\mu$L elution buffer. For RT-PCR, 5 $\mu$L of the RNA template was tested using real-time RT-PCR primer/probe sets for 2019-nCoV_N2 [10]. PCR was conducted using the Thunderbird probe one-step qRT-PCR kit by TOYOBO and QuantStudio 6 Pro real-time PCR system by Thermo Fisher Scientific following protocol by National Institute for Infectious Diseases (NIID RT-PCR) (10). For the N2 kit, 5 $\mu$L of RNA template was mixed with 15 $\mu$L of reaction reagent, 19 $\mu$L of which was dispensed onto a reaction panel chip. Primers and probes were targeted to the N2 genes. The sensitivity, specificity, positive predictive value (PPV), and negative predictive value (NPV) of MVP-N2 kit protocol compared to MVP-NIID RT-PCR protocol were calculated.

Address correspondence to Kenji Ota, kenjiotamd@nagasaki-u.ac.jp.

*Present address: Kazuko Yamamoto, First Department of Internal Medicine, Division of Infectious, Respiratory and Digestive Medicine, University of the Ryukyus Graduate School of Medicine, Nakagami, Okinawa, Japan, and Clinical Research Center, National Hospital Organization Nagasaki Medical Center, Omura, Nagasaki, Japan.

The authors declare a conflict of interest. This study was funded by Kyorin Pharmaceutical Co., Ltd., Tokyo, Japan. This work was supported by MEXT KAKENHI Grant Number JP21418576, Grant-in-Aid for Early-Career Scientists.

**TABLE 1** Evaluation of GeneSoC N2 detection kit on clinical saliva sample[a]

| MVP-N2 kit | MVP-NIID RT-PCR | | |
| --- | --- | --- | --- |
| | **Positive** | **Negative** | **Total** |
| Positive | 50 | 0 | 50 |
| Negative | 0 | 50 | 50 |
| Total | 50 | 50 | 100 |

[a]For the evaluation of GeneSoC N2 detection kit (N2 kit), sample preparation was performed by MagMAX Viral Pathogen (MVP, nucleic acid extraction and purification). The performance of N2 kit was compared to RT-PCR by National Institute for Infectious Diseases (NIID). Sensitivity 100.0% (50 / 50), Specificity 100.0% (50 / 50), PPV 100.0% (50 / 50), NPV 100.0% (50 / 50).

To evaluate the performance of the GLB, nucleic acid was extracted using MVP (both nucleic acid extraction and purification) or GLB (only nucleic acid extraction). For the GLB, saliva samples and buffer were mixed in a 3:15 ratio and left for 5 min at room temperature. Following each process, 5 $\mu$L of each sample was mixed with 15 $\mu$L of reaction reagent, 19 $\mu$L of which was used in GeneSoC N2 kit for nucleic acid amplification and detection. The sensitivity, specificity, PPV, and NPV of GLB-N2 kit protocol compared to MVP-N2 kit protocol were calculated.

The results of 100 saliva samples obtained from COVID-19 patients and healthy controls (50 samples each) are presented in Tables 1 and 2. In the evaluation of the N2 kit, perfect concordance with conventional RT-PCR was observed (Table 1). In the evaluation of GLB, five samples showed negative results by GLB among the 50 positive result samples by MVP (Table 2). The cycle threshold (Ct, mean $\pm$ standard deviation) values determined by GeneSoC were significantly higher in the GLB groups compared to the MVP groups (39.44 $\pm$ 6.64 versus 31.18 $\pm$ 5.65, $P < 0.0001$, Fig. S1). As described above, the sample was condensed four times in MVP and six times in GLB, leading to 24-fold changes (=2 ^ 4.58...) of nucleic acid concentration, which partially explains the difference of Ct value between these methods. The rest is considered to be derived from the methodological difference of sample preparation (the presence or absence of nucleic acid purification) and nucleic acid amplification. For negative samples by MVP, one sample from the healthy controls was excluded because of undetermined results with atypical amplification curves. The sensitivity was 90%, specificity was 100%, PPV was 100.0%, and NPV was 90.7%.

In this study, the N2 kit showed satisfactory performance in detection of SARS-CoV-2 RNA from saliva. Considering its short TAT, it has potential to replace conventional RT-PCR. The GLB showed slightly decreased sensitivity compared to conventional method. However, false-negative results were observed in samples containing low viral loads with high Ct values, where little viable virus was expected (11). Therefore, this result supports the clinical usefulness of the GLB for detecting infectious samples in saliva. In clinical settings, repeated test should be warranted in suspicious case, since increasing virus titer can follow negative test result in an initial phase of the infection. In those cases, GeneSoC system can be beneficial with its short TAT and relatively simple procedure. Another advantage of GLB is that the cost is not affected by test number. The cost per sample in MVP is relatively high when the small number of samples are tested in single run, while the cost gets lower when the sample number increases. In GLB, sample preparation is performed for each single sample, respectively, so the cost does

**TABLE 2** Evaluation of GeneSoC Lysis buffer on clinical saliva sample[a]

| GLB-N2 kit | MVP-N2 kit | | |
| --- | --- | --- | --- |
| | **Positive** | **Negative** | **Total** |
| Positive | 45 | 0 | 45 |
| Negative | 5 | 49 | 54 |
| Total | 50 | 49 | 99 |

[a]For the evaluation of GeneSoC lysis buffer (GLB, nucleic acid extraction) compared to MVP kit, nucleic acid amplification and detection was performed by GeneSoc N2 kit. Sensitivity 90% (45 / 50), Specificity 100.0% (49 / 49), PPV 100.0% (45 / 45), NPV 90.7% (49 / 54). PPV, Positive Predictive Value; NPV, Negative Predictive Value.

not elevate by sample number. Therefore, GLB shows its handiness and usefulness at its best in the setting where huge sample number is not required.

As a result, these kits showed potential for use in clinical settings. The application of saliva samples to the GeneSoC system can be beneficial for the detection of SARS-CoV-2 and COVID-19 diagnosis.

## SUPPLEMENTAL MATERIAL

Supplemental material is available online only.

**SUPPLEMENTAL FILE 1**, PDF file, 0.04 MB.

## ACKNOWLEDGMENT

This study was funded by Kyorin Pharmaceutical Co., Ltd., Tokyo, Japan. This work was also supported by MEXT KAKENHI Grant Number JP21418576, Grant-in-Aid for Early-Career Scientists.

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
