## [Reviewer comments · Microbiology Spectrum]

Microbiology Spectrum

Performance of the GeneSoC rapid PCR system in detection of SARS-CoV-2 from saliva specimens

Kenji Ota, Ryo Kurahara, Chie Tsukamoto, Yasuhide Kawamoto, Norihiko Akamatsu, Daisuke Sasaki, Fujiko Mitsumoto-Kaseida, Kei Sakamoto, Kosuke Kosai, Hiroo Hasegawa, Kazuko Yamamoto, KOICHI IZUMIKAWA, Hiroshi Mukae, and Katsunori Yanagihara

Corresponding Author(s): Kenji Ota, Nagasaki Daigaku Byoin

Review Timeline:

Submission Date:	August 23, 2022
Editorial Decision:	September 19, 2022
Revision Received:	October 28, 2022
Editorial Decision:	November 8, 2022
Revision Received:	January 13, 2023
Accepted:	January 16, 2023

Editor: Tulip Jhaveri

Reviewer(s): The reviewers have opted to remain anonymous.

Transaction Report:

DOI: <https://doi.org/10.1128/spectrum.03259-22>

September 19, 2022

Dr. Kenji Ota
Nagasaki University Hospital
Laboratory Medicine
Sakamoto, 1-7-1
Nagasaki
Japan

Re: Spectrum03259-22 (Performance of the GeneSoC rapid PCR system in detection of SARS-CoV-2 from saliva specimens)

Dear Dr. Kenji Ota:

Thank you for submitting your manuscript to Microbiology Spectrum. I am willing to consider moving forward with the manuscript, however it would need major revisions as outlined by the reviewers before we accept it. When submitting the revised version of your paper, please provide (1) point-by-point responses to the issues raised by the reviewers as file type "Response to Reviewers," not in your cover letter, and (2) a PDF file that indicates the changes from the original submission (by highlighting or underlining the changes) as file type "Marked Up Manuscript - For Review Only". Please use this link to submit your revised manuscript - we strongly recommend that you submit your paper within the next 60 days or reach out to me. Detailed instructions on submitting your revised paper are below.

Link Not Available

Sincerely,

Tulip Jhaveri

Journals Department
Reviewer comments:

Reviewer #1 (Comments for the Author):

The authors present a concise, well written report.

Major comments:

Does the lysis buffer have a more exact name? There are many lysis buffers out there so this does not readily identify what has been used.

Stats on the comparisons between the two approaches should be included.

It might be helpful to show a graph comparing the Ct values of the RNA extraction and lysis buffer preparation, with a line connecting the results for the same sample tested in each. This will help to inform the reader of the effect of lysis buffer preparation vs. RNA extraction.

Minor comment:

Line 46: citation for the protocol used

Reviewer #2 (Comments for the Author):

The manuscript "Performance of the GeneSoC rapid PCR system in detection of SARS-CoV-2 from saliva specimens" is a short paper aiming at assess the use of GeneSoc and a Lysis buffer to detect SARS-CoV-2 in saliva. The manuscript has no sections at all, difficulting its readability. The manuscript is too simplistic and lacks relevant information. Considering that its main goal is to demonstrate the adequacy of saliva as a specimen to detect SARS-CoV-2 using GeneSoc, the introduction (probably between line 23 and 40) could be more focused in this point and present also published evidences of why the saliva has not been used so widely as the convenience and advantages of its use might suppose. In fact, saliva is a convenient and easy to obtain specimen but there are many factors that may influence the results when it is used as a specimen to detect genetic material. The collection, transport, storage and processing of saliva can be critical for the success of the detection, and the authors give very scarce (or almost none) details on these procedures. As far as it possible to understand from the not very informative text about the methods, authors tried to compare the results obtained when extracted and purified RNA was used vs RNA obtained and present in a lysis solution which contained also saliva - however it is not clear which protocol was used to amplify the viral genes. Authors do not present statistical data. Repeatability, reproducibility and sensitivity were not determined.

Some specific comments:

Lines 41-42 - RNA was eluted?

Line 55 - Table?

Line 67 - "...where no viable virus was expected" - Why not? And if the Ct values are high only because the patients are in an initial phase of the incubation?

Staff Comments:

Preparing Revision Guidelines

Please return the manuscript within 60 days; if you cannot complete the modification within this time period, please contact me. If you do not wish to modify the manuscript and prefer to submit it to another journal, please notify me of your decision immediately so that the manuscript may be formally withdrawn from consideration by Microbiology Spectrum.

Response to Reviewers

We really appreciate the Reviewers for valuable comments.
Point-by-point responses to the raised issues are described as follows.
Thank you very much for reviewing our revised manuscript.

Reviewer #1

Major comments #1

Does the lysis buffer have a more exact name? There are many lysis buffers out there so this does not readily identify what has been used.

Response #1

Thank you very much for pointing out the name for the Lysis Buffer. For the readability, we changed "Lysis Buffer" to "GeneSoC Lysis Buffer (GLB)".

Major comments #2

Stats on the comparisons between the two approaches should be included.

Response #2

Thank you for the comment about statistical analysis. In comparisons of the two approaches, we calculated sensitivity, specificity, positive predictive value, and negative predictive value. We added the description as follows.

Line 56-58; "The sensitivity (Se), specificity (Sp), positive predictive value (PPV), and negative predictive value (NPV) of MVP-N2 kit protocol compared to MVP-RT-PCR protocol were calculated."

Line 71-72; "The Se, Sp, PPV, and NPV of GLB-N2 kit compared to MVP-N2 kit were calculated."

Major comments #3

It might be helpful to show a graph comparing the Ct values of the RNA extraction and lysis buffer preparation, with a line connecting the results for the same sample tested in each. This will help to inform the reader of the effect of lysis buffer preparation vs. RNA extraction.

Response #3

Thank you for the thoughtful suggestion. As the Reviewer #1's comment, we added a graph (Supplementary Figure 1) comparing the Ct values and its description. We believe this figure is informative in showing the difference of saliva sample preparation.

Line 80-83; "The cycle threshold (Ct, mean \pm standard deviation) values determined by GeneSoC were significantly higher in the GLB groups compared to MVP groups (39.44 ± 6.64 vs. 31.18 ± 5.65 , $P < 0.0001$, Supplementary Figure), partly due to decreased tested sample volume in GeneSoC."

Line 151-155; "Supplementary Figure. The cycle threshold (Ct) values determined by GeneSoC are shown, with lines connecting the results for the same sample tested in each. Five samples with negative results by GLB are expressed as Ct = 50. Results were compared with the use of a Paired t test. ****, $P < 0.0001$. GLB, GeneSoC Lysis Buffer; MVP, MagMAX Viral Pathogen."

Minor comment #1

Line 46: citation for the protocol used

Response #1

Thank you for pointing out. We added the citation #10.

Reviewer #2

Comment #1

The manuscript has no sections at all, difficulting its readability.

Response #1

Thank you very much for your time and effort for reviewing our manuscript. As this is a “new data letter”, we have made no sections in the manuscript.

Comment #2

The manuscript is too simplistic and lacks relevant information. Considering that its main goal is to demonstrate the adequacy of saliva as a specimen to detect SARS-CoV-2 using GeneSoc, the introduction (probably between line 23 and 40) could be more focused in this point and present also published evidences of why the saliva has not been used so widely as the convenience and advantages of its use might suppose. In fact, saliva is a convenient and easy to obtain specimen but there are many factors that may influence the results when it is used as a specimen to detect genetic material. The collection, transport, storage and processing of saliva can be critical for the success of the detection, and the authors give very scarce (or almost none) details on these procedures.

Response #2

We really appreciate these comments regarding the characteristics and issues about saliva specimens for the detection of SARS-CoV-2. We agree that the collection, transport, storage and processing of saliva are important factors in applying saliva for clinical specimen and need to be standardized. Therefore, we consider that individual evaluation of test kit is important in saliva specimen.

We mentioned the descriptions below and added details about how the saliva samples were handled in this study.

Line 33-36; “However, collection, storage, and processing methods for saliva specimens are not standardized sufficiently [9], so the study comparing saliva and other types of specimens needs careful interpretation before clinical application. In addition, sample preparation including nucleic acid extraction is another time-consuming process for using saliva as test specimen. Therefore, individual evaluation for SARS-CoV-2 detection method in saliva is necessary.”

Line 73-75; “Clinically obtained and stored (-80 °C) saliva samples were used in this study. Saliva specimens were self-collected by spitting directly into a sterile tube, with fasting prior 30 minutes. The samples had been stored at 4 °C before transported to our laboratory.”

Comment #3

As far as it possible to understand from the not very informative text about the methods, authors tried to compare the results obtained when extracted and purified RNA was used vs RNA obtained and present in a lysis solution which contained also saliva - however it is not clear which protocol was used to amplify the viral genes.

Response #3

We appreciate the Reviewer #3 for this comment. For the clarity, we modified the descriptions as follows.

Line 56-58; “The sensitivity (Se), specificity (Sp), positive predictive value (PPV), and negative predictive value (NPV) of MVP-N2 kit protocol compared to MVP-RT-PCR protocol were calculated.”

Line 71-72; “The Se, Sp, PPV, and NPV of GLB-N2 kit protocol compared to MVP-N2 kit protocol were calculated.”

Comment #4

Authors do not present statistical data.

Response #4

Thank you very much for mentioning important point. Relevant to Comment #2 from the Reviewer #1, we added statistical descriptions.

Comment #5

Repeatability, reproducibility and sensitivity were not determined.

Response #5

We appreciate the comments about repeatability, reproducibility and sensitivity. As these

issues are limited by this study design, we focused on the performance of GeneSoC Lysis Buffer and N2 kit on saliva specimens, using a total of 100 clinical obtained samples. The assessment for repeatability, reproducibility of GeneSoC system for nucleic acid amplification and detection are yielded to other research. Similarly, we have not tried to determine the limit of detection in this study.

Some specific comments:

Comment #1.

Lines 41-42 - RNA was eluted?

Response #1.

Thank you very much for pointing out our inaccurate description about methods. We have made change as follows.

Line 49-50; "Briefly, nucleic acid in saliva specimens (400 μ L) were purified and condensed in 100 μ L elution buffer."

Comment #2.

Line 55 - Table?

Response #2.

As the Reviewer #2's comment, we have changed "Table" to "Table A and B".

Comment #3.

Line 67 - "...where no viable virus was expected" - Why not? And if the Ct values are high only because the patients are in an initial phase of the incubation?

Response #3.

We really appreciate Reviewer #2 for providing us the chance to have discussion about this important theme regarding viral dynamics. We agree that single negative result sometimes indicates an initial phase of infection. Therefore, repeated test should be performed in suspicious cases. In those cases, the advantages of GeneSoC system with simple procedure and short TAT are sure to be beneficial. We added the descriptions as follows.

Line 92-98; “However, false-negative results were observed in samples containing low viral loads with high Ct values, where few viable virus was expected [11]. Therefore, this result supports the clinical usefulness of the GLB for detecting infectious samples in saliva. In clinical settings, repeated test should be warranted in suspicious case, since increasing virus titer can follow negative test result in an initial phase of the infection. In those cases, GeneSoC system can be beneficial with its short TAT and relatively simple procedure.”

November 8, 2022

Dr. Kenji Ota
Nagasaki University Hospital
Laboratory Medicine
Sakamoto, 1-7-1
Nagasaki
Japan

Re: Spectrum03259-22R1 (Performance of the GeneSoC rapid PCR system in detection of SARS-CoV-2 from saliva specimens)

Dear Dr. Kenji Ota:

Link Not Available

Sincerely,

Tulip Jhaveri

Journals Department
Reviewer comments:

Reviewer #1 (Comments for the Author):

Thank you to the authors for making the suggested edits to make their manuscript stronger.

I have a few more comments:

As variants change, I think it would be important to state during which period this study was conducted - what months the samples were collected - to infer the possible variants being detected. It would be good to report on the in silico results of the N2 target and whether it has remained unaffected with emerging variants or whether some variants exhibit nucleotide changes in

the targeted region.

As Se and Sp have the same word count as sensitivity and specificity I suggest writing these out in full as Se and Sp are not commonly used to represent these (unlike PPV and NPV being better known).

Lines 40-41: I suggest stating that it is to investigate the performance of GLB and the N2 kit - in that, you first do the lysis with the GLB and then test in PCR with the N2 kit.

Lines 61-63, about saliva collection, should come perhaps before line 43 - the first mention of testing saliva.

Lines 70-71: the reader would benefit from knowing more about what you mean about decreased sample testing volume - ie, a greater amount is extracted via MVP and concentrated into X ul elution buffer creating a concentration factor of X times. Is the difference in Ct then aligned to that concentration factor?

Is there a price benefit using GeneSoc vs. MVP? This may be another factor which could offset the lower sensitivity.

Did you run an LOD study on the GeneSoc and test side by side in the other PCR or via MVP? This would also be a clearer comparison and what is required by regulatory agencies around the world when validating new tests.

For the tables, it is not clear to the reader what exactly is being compared and why. More information should be included for A, stating what 'RT-PCR' assay the N2 detection kit is being compared to. For Table B, the reader would benefit from being able to easily understand what 'MVP' is and how this approach differs to the lysis buffer approach (ie, extraction vs. extraction-free?).

For the supplementary figure, I suggest adding 'kit' after MagMax Viral Pathogen.

Reviewer #2 (Comments for the Author):

In general, the authors have revised the manuscript taking into account the comments made to the manuscript previously submitted.

Staff Comments:

Preparing Revision Guidelines

Please return the manuscript within 60 days; if you cannot complete the modification within this time period, please contact me. If you do not wish to modify the manuscript and prefer to submit it to another journal, please notify me of your decision immediately so that the manuscript may be formally withdrawn from consideration by Microbiology Spectrum.

Response to Reviewers

We really appreciate the Reviewers for suggesting a series of improvement in your valuable comments.

We sincerely respond to your comments and revised our manuscript as follows. We appreciate your time and effort for reviewing our manuscript again.

Reviewer #1

Comments #1

As variants change, I think it would be important to state during which period this study was conducted - what months the samples were collected - to infer the possible variants being detected.

Response #1

Thank you very much for the thoughtful comment. We agree the importance of stating in which period the samples were obtained. In order to validate the performance in recent variants, we selected the samples obtained from February 2022 to March 2022, when omicron variants were dominant. We added following description,

Line 43-45, "These samples were obtained from February 2022 to March 2022, when omicron variants were dominant."

Comments #2

It would be good to report on the in silico results of the N2 target and whether it has remained unaffected with emerging variants or whether some variants exhibit nucleotide changes in the targeted region.

Response #2

Thank you very much for this comment regarding the relationships between PCR target and mutations. We have not performed in silico analysis in this study. The target of GeneSoC N2 kit is designed to detect and amplify the same region as the protocol by National Institute for Infectious Diseases. This is another merit of N2 kit, since detection failure due to nucleotide changes can be theoretically screened by our conventional method.

Comments #3

As Se and Sp have the same word count as sensitivity and specificity I suggest writing these out in full as Se and Sp are not commonly used to represent these (unlike PPV and NPV being better known).

Response #3

Thank you for the suggestion. We changed Se and Sp to sensitivity and specificity, respectively.

Comments #4

Lines 40-41: I suggest stating that it is to investigate the performance of GLB and the N2 kit - in that, you first do the lysis with the GLB and then test in PCR with the N2 kit.

Response #4

Thank you for the comment for the comprehensibility. As per the suggestion, we modified the description.

Comments #5

Lines 61-63, about saliva collection, should come perhaps before line 43 - the first mention of testing saliva.

Response #5

Thank you for the comment for the composition. As per the suggestion, we changed the manuscript.

Comments #6

Lines 70-71: the reader would benefit from knowing more about what you mean about decreased sample testing volume - ie, a greater amount is extracted via MVP and concentrated into X ul elution buffer creating a concentration factor of X times. Is the difference in Ct then aligned to that concentration factor?

Response #6

Thank you very much for this very important comment.

In MVP, 400 μ L of saliva were condensed into 100 μ L elution buffer (4x concentration). In GLB, saliva and buffer were mixed in a 3:15 ratio (1/6x concentration). Therefore, 24 ($=2^4 \cdot 4.58\dots$) fold change of sample concentration theoretically occurs between these two protocols. This partially explains the difference of Ct in this study (8.26, 39.44 vs. 31.18), the rest is considered to be derived from the methodological difference of sample preparation (presence or absence of nucleic acid purification). We described this point as follows,

Line 72-76, "As described above, sample was condensed four times in MVP and one sixths in GLB, leading to 24 fold changes ($=2^4 \cdot 4.58\dots$) of nucleic acid concentration, which partially explains the difference of Ct value between MVP and GLB. The rest is considered to be derived from the methodological difference such as the presence or absence of nucleic acid purification."

Comments #7

Is there a price benefit using GeneSoc vs. MVP? This may be another factor which could offset the lower sensitivity.

Response #7

Thank you for this viewpoint. We agree that the economic factor is an important factor in evaluating test protocol. When testing one sample in a run, GLB costs 1,188 Japanese yen per sample, whereas MVP costs 4,019 Japanese yen per sample (including chip comb and deep well plate), suggesting GLB's price benefit. The price of MVP per sample gets lower as the sample number per run increases, while the price of GLB does not change according to the sample number. Therefore, the handiness and cost stability regardless of sample size is another benefit of GLB. We appreciate the Reviewer for providing us with a chance to consider the economic point. We added the description as follows,

Line 88-93, "Another advantage of GLB is that the cost is not affected by test number. The cost per sample in MVP is relatively high when the small number of samples are tested in single run, while the cost gets lower when the sample number increases. In GLB, sample preparation is performed for each single sample respectively, so the cost does not elevate by sample number. Therefore, GLB shows its handiness and usefulness at its best in the setting where huge sample number is not required."

Comments #8

Did you run an LOD study on the GeneSoc and test side by side in the other PCR or via MVP? This would also be a clearer comparison and what is required by regulatory agencies around the world when validating new tests.

Response #8

Thank you very much for this important comment. Since the aim of this study is to evaluate the GLB and N2 kit using saliva samples, we have not performed an LOD study. We totally agree the Reviewer's comment and would like to plan LOD study comparing other methods and samples.

Comments #9

For the tables, it is not clear to the reader what exactly is being compared and why. More information should be included for A, stating what 'RT-PCR' assay the N2 detection kit is being compared to. For Table B, the reader would benefit from being able to easily understand what 'MVP' is and how this approach differs to the lysis buffer approach (ie, extraction vs. extraction-free?).

Response #9

Thank you very much for the comment about Tables. We modified the tables and legends for the clarity.

Comments #10

For the supplementary figure, I suggest adding 'kit' after MagMax Viral Pathogen.

Response #10

Thank you very much for the comment about supplementary Figures. We modified the supplementary figure legends as per suggestion.

January 16, 2023

Dr. Kenji Ota
Nagasaki Daigaku Byoin
Laboratory Medicine
Sakamoto, 1-7-1
Nagasaki
Japan

Re: Spectrum03259-22R2 (Performance of the GeneSoC rapid PCR system in detection of SARS-CoV-2 from saliva specimens)

Dear Dr. Kenji Ota:

Your manuscript has been accepted, and I am forwarding it to the ASM Journals Department for publication. You will be notified when your proofs are ready to be viewed.

Sincerely,

Tulip Jhaveri
Editor, Microbiology Spectrum
